# NEURAL CODE COMPLETION

**Chang Liu**[*]**, Xin Wang**[*]**, Richard Shin, Joseph E. Gonzalez, Dawn Song**
University of California, Berkeley

## ABSTRACT

Code completion, an essential part of modern software development, yet can be challenging for dynamically typed programming languages. In this paper we explore the use of neural network techniques to automatically learn code completion from a large corpus of dynamically typed JavaScript code. We show different neural networks that leverage not only token level information but also structural information, and evaluate their performance on different prediction tasks. We demonstrate that our models can outperform the state-of-the-art approach, which is based on decision tree techniques, on both next non-terminal and next terminal prediction tasks by 3.8 points and 0.5 points respectively. We believe that neural network techniques can play a transformative role in helping software developers manage the growing complexity of software systems, and we see this work as a first step in that direction.

## 1 INTRODUCTION

As the scale and complexity of modern software libraries and tools continue to grow, code completion has become an essential feature in modern integrated development environments (IDEs). By suggesting the right libraries, APIs, and even variables in real-time, intelligent code completion engines can substantially accelerate software development. Furthermore, as many projects move to dynamically typed and interpreted languages, effective code completion can help to reduce costly errors by eliminating typos and identifying the right arguments from context.

However, existing approaches to intelligent code completion either rely on strong typing (e.g., Visual Studio for C++), which limits their applicability to widely used dynamically typed languages (e.g., JavaScript and Python), or are based on simple heuristics and term frequency statistics which are often brittle and are relatively error-prone. In particular, Raychev et al. (2016a) proposes the state-of-the-art probabilistic model for code, which generalizes both simple $n$-gram models and probabilistic grammar approaches. This approach, however, examines only a limited number of elements in the source code when completing the code. Therefore, the effectiveness of this approach may not scale well to large programs.

In this paper we explore the use of deep learning techniques to address the challenges of code completion for the widely used and dynamically typed JavaScript programming language. We formulate the code completion problem as a sequential prediction task over the traversal of a parse-tree structure consisting of both non-terminal structural nodes and terminal nodes encoding program text. We then present simple, yet expressive, LSTM-based (Hochreiter & Schmidhuber (1997)) models that leverage additional side information obtained by parsing the program structure.

Compared to widely used heuristic techniques, deep learning for code completion offers the opportunity to learn rich contextual models that can capture language and even library specific code patterns without requiring complex rules or expert intervention.

We evaluate our recurrent neural network architecture on an established benchmark dataset for the JavaScript code completion. Our evaluations reveal several findings: (1) when evaluated on short programs, our RNN-based models can achieve better performance on the next node prediction tasks compared to the prior art (Bielik et al. (2016); Raychev et al. (2016a)), which are based on decision-tree models; (2) our models' prediction accuracies on longer programs, which is provided in the test set, but were not evaluated upon by previous work, are better than our models' accuracies on shorter

---

[*]The first and second authors contributed equally and are listed in an alphabetical order.

Figure 1: Code Completion Example in IntelliJ IDEA

Figure 2: Correct prediction of the program in Figure 1

programs; and (3) in the scenario that the code completion engine suggests a list of candidates, our RNN-based models allow users to choose from a list of 5 candidates rather than inputting manually for over 96% of all time when this is possible.

These promising results encourage more investigation into developing neural network approaches for the code completion problem. We believe that our work not only highlights the importance of the field of neural network-based code completion, but is also an important step toward neural network-based program synthesis.

## 2 RELATED WORK

Existing approaches that build probabilistic models for code can typically be categorized as $n$-gram models (Hindle et al., 2012; Nguyen et al., 2013; Tu et al., 2014), probabilistic grammars (Collins, 2003; Allamanis & Sutton, 2014; Allamanis et al., 2015; Maddison & Tarlow, 2014; Liang et al., 2010), and log-bilinear models (Allamanis et al., 2015). Bielik et al. (2016) generalizes the PCFG approach and n-gram approach, while Raychev et al. (2016a) further introduces decision tree approaches to generalize Bielik et al. (2016).

Raychev et al. (2014) and White et al. (2015) explore how to use recurrent neural networks (RNNs) to facilitate the code completion task. However, these works only consider running RNNs on top of a token sequence to build a probabilistic model. Although the input sequence considered in Raychev et al. (2014) is produced from an abstract object, the structural information contained in the abstract syntax tree is not directly leveraged by the RNN structure in both of these two works. In contrast, we consider extending LSTM, a RNN structure, to leverage the structural information directly for the code prediction task.

Recently there has been an increasing interest in developing neural networks for program synthesis (Ling et al. (2016); Beltagy & Quirk (2016); Dong & Lapata (2016); Chen et al. (2016)). These works all consider synthesizing a program based on inputs in other formats such as images or natural language descriptions.

## 3 CODE COMPLETION VIA BIG CODE

In this section, we first introduce the problem of code completion and its challenges. Then we explain abstract syntax trees (AST), which we use as the input for our problems. Lastly, we formally define the code completion problem in different settings as several prediction problems based on a partial AST.

### 3.1 CODE COMPLETION: AN EXAMPLE

Code completion is a feature in some integrated development environments (IDEs) to speed up programmers' coding process. Figure 1 demonstrates this feature in IntelliJ IDEA[1]. In this example,

---

[1] https://www.jetbrains.com/idea/

a part of a JavaScript program has been input to the IDE. When the dot symbol (i.e., ".") is added after `__webpack_require__`, the IDE prompts with a list of candidates that the programmer is most likely to input next. When a candidate matches the intention, the programmer can choose it from the list rather than typing it manually. In this work, we define the code completion problem as predicting the next symbol while a program is being written. We consider this problem as an important first step toward completing an entire program.

Traditional code completion techniques are developed by the programming language community to leverage context information for prediction. For example, when a programmer writes a Java program and inputs a variable name and then a dot symbol, the code completion engine will analyze the class of the variable and prompt the members of the class. In programming language literature, such information is referred to as *type information*. Statically typed languages, such as C and Java, enforces type checking at static time, so that the code completion engine can take advantage of full type information to make prediction without executing the code.

In recent years, dynamically typed languages, such as Python or JavaScript, have become increasingly popular. In these languages, type checking is usually performed dynamically while executing a program. Thus, type information may be only partially available to the code completion engine while the programmer is writing the code. Despite their popularity, the dynamic typing of these languages makes code completion for them challenging. For example, in Figure 1, the next symbol to be added is `p`. This symbol does not appear in the previous part of the program, and thus the code completion engine in IntelliJ IDEA IDE cannot prompt with this symbol.

However, this challenge may be remedied by leveraging a large corpus of code, a.k.a., *big code*. In fact, `__webpack_require__.p` is a frequently used combination appearing in many programs on `Github.com`, one of the largest repositories of source code. Therefore, a code completion engine powered by big code is likely to learn this combination and to prompt `p`. In fact, our methods discussed in later sections can predict this case very well (Figure 2),

## 3.2 ABSTRACT SYNTAX TREE

Regardless of whether it is dynamically typed or statically typed, any programming language has an unambiguous context free grammar (CFG), which can be used to parse source code into an *abstract syntax tree (AST)*. Further, an AST can be converted back into source code easily. Therefore we consider the input of our code completion problem as an AST, which is a typical assumption made by most code completion engines.

An AST is a rooted tree. In an AST, each non-leaf node corresponds to a *non-terminal* in the CFG specifying structure information. In JavaScript, non-terminals may be `ExpressionStatement`, `ForStatement`, `IfStatement`, `SwitchStatement`, etc. Each leaf node corresponds to a *terminal* in the CFG encoding program text. There are infinite possibilities for terminals. They can be variable names, string or numerical literals, operators, etc.

Figure 3 illustrates a part of the AST of the code snippet in Figure 1. In this tree, a node without a surrounding box (e.g., `ExpressionStatement`, etc.) denotes a non-terminal node. A node embraced by an orange surrounding box (e.g., `installedModules`) denotes a terminal node. At the bottom of the figure, there is a non-terminal node `Property` and a terminal node `p`. They have not been observed by the editor, so we use green to indicate this fact. Note that each non-terminal has at most one terminal as its child.

In a traditional code completion engine, the AST can be further processed by a type checker so that type information will be attached to each node. In this work, however, we focus on dynamically typed languages, and type information is not always available. Therefore, we do not consider the type information provided by a compiler, and leave it for our future work.

## 3.3 PROBLEM SETUP

In this work, we consider the input to be a *partial AST*, and the code completion problem is to predict the next node given the partial AST. In the following, we first define a partial AST, and then present the code completion problems in different scenarios.

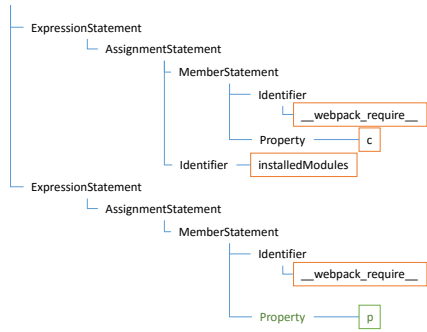

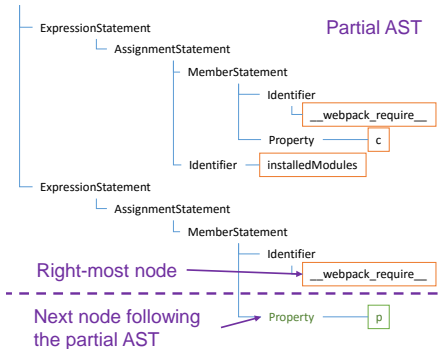

Figure 3: AST example (part)          Figure 4: Partial AST example

**Input: a partial AST.**  Given a complete AST $T$, we define *a partial AST* to be a subtree $T'$ of $T$, such that for each node $n$ in $T'$, its left set $L_T(n)$ with respect to $T$ is a subset of $T'$, i.e., $L_T(n) \subseteq T'$. Here, the left set $L_T(n)$ of a node $n$ with respect to $T$ is defined as the set of all nodes in the in-order sequence during the depth-first search of $T$ that are visited earlier than $n$ .

Under this definition, in each partial AST $T'$, there exists the *right-most node* $n_R$, such that all other nodes in $T'$ form its left set $L_T(n_R)$. The next node in the in-order depth-first search visiting sequence after $n_R$ is also the first node not appearing in $T'$. We call this node *the next node following the partial AST*. Figure 4 illustrates these concepts using the example in Figure 3. In the rest of the paper, we also refer to a partial AST as a *query*.

**Next node prediction.**  Given a partial AST, the *next node prediction problem*, as suggested by its name, is to predict the next node following the partial AST. Based on the node's kind, i.e., whether its a non-terminal node or a terminal one, we can categorize the problem into the next non-terminal prediction problem and the next terminal prediction problem. Although the next terminal prediction problem may sound more interesting, the next non-terminal prediction problem is also important, since it predicts the structure of the program. For example, when then next non-terminal is `ForStatement`, the next token in the source program is the keyword **for**, which does not have a corresponding terminal in the dataset. In this case, a model able to predict the next non-terminal can be used by the code-completion engine to emit the keyword **for**. These two tasks are also the same problems considered by previous works employing domain specific languages to achieve heuristic-based code completion (Raychev et al. (2016b); Bielik et al. (2016)).

**Predicting the next node versus predicting the next token.**  A natural alternative formulation of the problem is predicting the next token given the token sequence that has been inputted so far. Such a formulation, however, does not take advantage of the AST information, which is very easy to acquire with a suitable parser. Predicting the next node allows taking advantage of such information to enable more intelligent code completion.

In particular, predicting the next non-terminal allows completing the structure of a code block rather than a single (keyword) token. For example, when the next token is a keyword `for`, the corresponding next non-terminal is `ForStatement`, which corresponding to the following code block:

```
for( ____ ; ____ ; ____ ) {
    // for-loop body
}
```

In this case, successfully predicting the next non-terminal node allows completing not only the next key token `for`, but also tokens such as `(`, `;`, `)`, `{`, and `}`. Such structure completion enabled by predicting the next non-terminal is more compelling in modern IDEs.

Predicting the next terminal node allows completing identifiers, properties, literals, etc., which is similar to the next token prediction. However, predicting the next terminal node can leverage the information of the predicting node's non-terminal parent, indicating what is being predicted, i.e., an identifier, a property, or a literal, etc. For example, when completing the following expression:

```
__webpack_require_.
```

the code completion engine with AST information will predict a *property* of `__webpack_require_`, while the engine without AST information only learns two tokens `__webpack_require_` and a dot "`.`" and tries to predict the next token without any constraint. In our evaluation, we show that by leveraging the information from the non-terminal parent can significantly improve the performance.

In this work, we focus on the next node prediction task, and leave the comparison with next token prediction as our future work.

**Joint prediction.** A more important problem than predicting only the next non-terminal or terminal itself is to predict the next non-terminal and terminal together. We refer to this task to predict both next non-terminal and terminal as *the joint prediction problem*. We hope code completion can be used to generate the entire parsing tree in the end, and joint prediction is one step further toward this goal than next node prediction.

Formally, the joint prediction problem that we consider is that, given a partial AST whose following node is a non-terminal one, we want to predict both the next non-terminal and the next terminal.

There may be non-terminal nodes which do not have a terminal child (e.g., the `AssignmentStatement`). In this case, we artificially add an `EMPTY` terminal as its child. Note that this treatment is the same as in Bielik et al. (2016). We count it as a correct prediction if both the next non-terminal and terminal are predicted correctly.

**Denying prediction.** There may be infinite possibilities for terminals, so it is impossible to predict all terminals correctly. We consider an alternative scenario that, when it thinks that the programmer will input a rare terminal, the code completion engine should have the ability to identify this case, and deny predicting the next node(s).

In our problem, we build a vocabulary for frequent terminals. All terminals not in this vocabulary are considered as an `UNK` terminal. In this case, when it predicts `UNK` for the next terminal, the code completion model is considered as denying prediction. Since non-terminals' vocabulary size is very small, denying prediction is only considered for the next terminal prediction, but not for the next non-terminal prediction.

## 4 MODELS

In this section, we present the basic models considered in this work. In particular, given a partial AST as input, we first convert the AST into its left-child right-sibling representation, and serialize it as its in-order depth first search sequence. Thus, we consider the input for the next non-terminal prediction as a sequence of length $k$, i.e., $(N_1, T_1), (N_2, T_2), ..., (N_k, T_k)$. Here, for each $i$, $N_i$ is a non-terminal, and $T_i$ is the terminal child of $N_i$. For each non-terminal node $N_i$, we encode not only its kind, but also whether the non-terminal has at least one non-terminal child, and/or one right-sibling. In doing so, from an input sequence, we can reconstruct the original AST. This encoding is also employed by Raychev et al. (2016a). We refer to each element in the sequence (e.g., $(N_i, T_i)$) as a *token*. As mentioned above, a non-terminal without a terminal child is considered to have an `EMPTY` child.

This input sequence $(N_1, T_1), (N_2, T_2), ..., (N_k, T_k)$ is the only input for all problems except the next terminal prediction. For the next terminal prediction problem, besides the input sequence, we also have the information about the parent of the current predicting terminal, which is a non-terminal, i.e., $N_{k+1}$.

Throughout the rest of the discussion, we assume that both $N_i$ and $T_i$ employ one-hot encoding. The vocabulary sets of non-terminals and terminals are separate.

### 4.1 NEXT NON-TERMINAL PREDICTION

Given an input sequence, our first model predicts the next non-terminal. The architecture is illustrated in Figure 5. We refer to this model as NT2N, which stands for *using the sequence of*

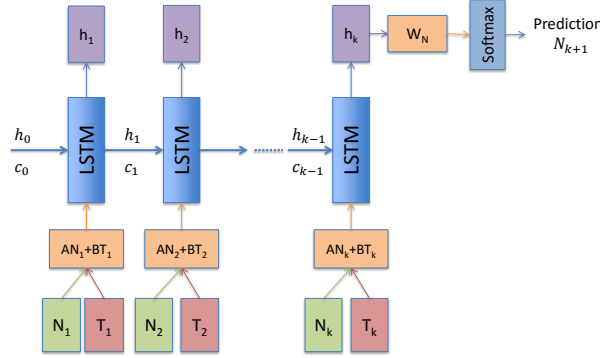

Figure 5: Architecture (NT2N) for predicting the next non-terminal.

*Non-terminal and Terminal pairs TO predict the next Non-terminal.* We first explain each layer of NT2N, and then introduce two variants of this model.

**Embedding non-terminal and terminal.** Given an input sequence, the embedding of each token is computed as

$$E_i = AN_i + BT_i \tag{1}$$

where $A$ is a $J \times V_N$ matrix and $B$ is a $J \times V_T$ matrix. Here $J$ is the size of the embedding vector, $V_N$ and $V_T$ are the vocabulary sizes of non-terminals and terminals respectively.

**LSTM layer.** Then the embedded sequence is fed into a LSTM layer to get the hidden state. In particular, a LSTM cell takes an input token and a hidden state $h_{i-1}, c_{i-1}$ from the previous LSTM cell as input, computes a hidden state $h_i, c_i$, and outputs $h_i$, based on the following formulas:

$$\begin{pmatrix} q \\ f \\ o \\ g \end{pmatrix} = \begin{pmatrix} \sigma \\ \sigma \\ \sigma \\ \tanh \end{pmatrix} \mathbf{P}_{J,2J} \begin{pmatrix} \mathbf{x}_i \\ \mathbf{h}_{i-1} \end{pmatrix}$$
$$\mathbf{c}_i = f \odot \mathbf{c}_{i-1} + q \odot g$$
$$\mathbf{h}_i = o \odot \tanh(\mathbf{c}_i)$$

Here, $\mathbf{P}_{J,2J}$ denotes a $J \times 2J$ parameter matrix, where $J$ is the size of the hidden state, i.e. dimension of $h_i$, which is equal to the size of embedding vectors. $\sigma$ and $\odot$ denote the sigmoid function and pointwise multiplication respectively.

**Softmax layer.** Assume $h_k$ is the output hidden state of the last LSTM cell. $h_k$ is fed into a softmax classifier to predict the next non-terminal. In particular, we have

$$\hat{N}_{k+1} = \mathbf{softmax}(W_N \times h_k + b_N)$$

where $W_N$ and $b_N$ are a matrix of size $V_N \times J$ and a $V_N$-dimensional vector respectively.

**Using only non-terminal inputs.** One variant of this model is to omit all terminal information from the input sequence. In this case, the embedding is computed as $E_i = AN_i$. We refer to this model as N2N, which stands for *using Non-terminal sequence TO predict the next Non-terminal*.

**Predicting the next terminal and non-terminal together.** Based on NT2N, we can predict not only the next non-terminal but also the next terminal, using

$$\hat{T}_{k+1} = \mathbf{softmax}(W_T \times h_k + b_T)$$

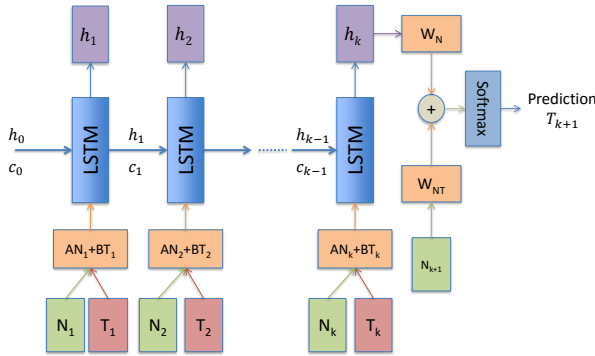

Figure 6: Architecture (NTN2T) for predicting the next terminal.

where $W_T$ and $b_T$ are a matrix of size $V_T \times J$ and a $V_T$-dimensional vector respectively. In this case, the loss function has an extra term to give supervision on predicting $\hat{T}$. We refer to this model as NT2NT, which stands for *using the sequence of Non-terminal and Terminal pairs TO predict the next Non-terminal and Terminal pair*.

## 4.2 NEXT TERMINAL PREDICTION

In the next terminal prediction problem, the partial AST does not only contain $(N_1, T_1), ..., (N_k, T_k)$, but also $N_{k+1}$. In this case, we can employ the architecture in Figure 6 to predict $T_{k+1}$. In particular, we first get the LSTM output $h_k$ in the same way as in NT2N. The final prediction is based on

$$\hat{T}_{k+1} = \mathbf{softmax}(W_T h_k + W_{NT} N_{k+1} + b_T)$$

where $W_{NT}$ is a matrix of size $V_T \times V_N$, and $W_T$ and $b_T$ are the same as in NT2NT. We refer to this model as NTN2T, which stands for *Non-terminal and Terminal pair sequence and the next Non-terminal TO predict the next Terminal*.

Note that the model NT2NT can also be used for the next terminal prediction task, although the non-terminal information $N_{k+1}$ is not leveraged. We will compare the two approaches later.

## 4.3 JOINT PREDICTION

We consider two approaches to predict the next non-terminal and the next terminal together. The first approach is NT2NT, which is designed to predict the two kinds of nodes together.

An alternative approach is to (1) use a next non-terminal approach $X$ to predict the next non-terminal; and (2) feed the predicted non-terminal and the input sequence into NTN2T to predict the next terminal. We refer to such an approach as $X$+NTN2T.

## 4.4 DENYING PREDICTION

We say a model *denies prediction* when it predicts the next terminal to be UNK, a special terminal to substitute rare terminals. However, due to the large amount of rare terminals, the occurrences of UNK may be much greater than any single frequent terminals. In this case, a model that can deny prediction may tend to predict UNKs, and thus may predict for fewer queries than it should.

To mitigate this problem, we modify the loss function to be adaptive. Specifically, training a machine learning model $f_\theta$ is to optimize the following objective:

$$\mathbf{argmin}_\theta \sum_i l(f_\theta(q_i), y_i)$$

where $\{(q_i, y_i)\}$ is the training dataset consisting pairs of a query $q_i$ and its ground truth next token $y_i$. $l$ is the loss function to measure the distance between the prediction $\hat{y}_i = f_\theta(q_i)$ and the ground

| Training set | | Test set | | Overall | |
|---|---|---|---|---|---|
| Programs | 100,000 | Programs | 50,000 | Non-terminal | 44 |
| Queries | $1.7 \times 10^8$ | Queries | $8.3 \times 10^7$ | Terminal | $3.1 \times 10^6$ |

Table 1: Statistics of the dataset

truth $y_i$. We choose $l$ to be the standard cross-entropy loss. We introduce a weight $\alpha_i$ for each sample $(q_i, y_i)$ in the training dataset to change the objective to be as follows:

$$\mathbf{argmin}_\theta \sum_i \alpha_i l(f_\theta(q_i), y_i)$$

When training a model not allowed to deny prediction, we set $\alpha_i = 0$ for $y_i = \texttt{UNK}$, and $\alpha_i = 1$ otherwise. In doing so, it is equivalent to remove all queries whose ground truth next token is $\texttt{UNK}$.

When training a model that allows denying prediction, we set all $\alpha_i$ to be 1. To denote this case, we put a notation "+D" at the end of the model, (e.g., NT2NT+D, etc.).

## 5 EVALUATION

### 5.1 DATASET

We use the JavaScript dataset[2] provided by Raychev et al. (2016b) to evaluate different approaches. The statistics of the dataset can be found in Table 1. Raychev et al. (2016a) provides an approach, called PHOG, for the next token prediction. The reported accuracy results are based on a subset of $5.3 \times 10^7$ queries from the full test set. Specifically, Raychev et al. (2016a) chose all queries in each program containing fewer than 30,000 tokens.[3] When we compare with their results, we use the same testset. Otherwise, without a special specification, our results are based on the full test set consisting of $8.3 \times 10^7$ queries.

### 5.2 TRAINING DETAILS

**Vocabulary**  In our dataset, there are 44 different kinds of non-terminals. Combining two more bits of information to indicate whether the non-terminal has a child and/or a right sibling, there are at most 176 different non-terminals. However, not all such combinations are possible: a `ForStatement` must have a child. In total, the vocabulary size for non-terminals is 97. For terminals, we sort all terminals in the training set by their frequencies. Then we choose the 50,000 most frequent terminals to build the vocabulary. We further add three special terminals: `UNK` for out-of-vocabulary tokens, `EOF` indicating the end of program, and `Empty` for the non-terminal which does not have a terminal. Note that about $45\%$ terminals in the dataset are `Empty` terminals.

**Training details.**  We use a single layer LSTM network with hidden unit size of 1500 as our base model. To train the model, we use Adam (Kingma & Ba (2014)) with base learning rate 0.001. The learning rate is multiplied by 0.9 every 0.2 epochs. We clip the gradients' norm to 5. The batch size is $b = 80$. We use truncated backpropagation through time, by unrolling the LSTM model $s = 50$ times to take an input sequence of length 50 in each batch (and therefore each batch contains $b \times s = 4000$ tokens).

We divide each program into segments consisting of $s$ consecutive tokens. The last segment of a program, which may not be full, is padded with $\langle \text{EOF} \rangle$ tokens. We coalesce multiple epochs together. We organize all training data into $b$ buckets. In each epoch, we randomly shuffle all programs in the training data to construct a queue. Whenever a bucket is empty, a program is popped from the queue and all segments of the program are inserted into the empty bucket sequentially. When the queue becomes empty, i.e., the current epoch finishes, all programs are re-shuffled randomly to reconstruct the queue. Each mini-batch is formed by $b$ segments, i.e., one segment popped from each bucket. When the training data has been shuffled for $e = 8$ times, i.e., $e$ epochs are inserted into the

---

[2]http://www.srl.inf.ethz.ch/js150
[3]This detail was not explained in the paper. We contacted the authors to confirm it.

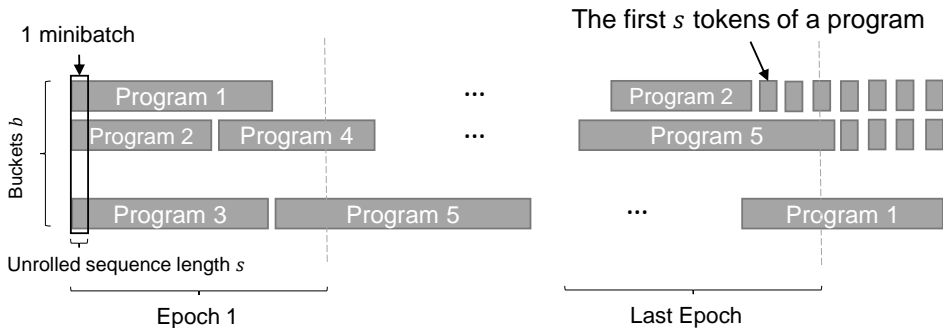

Figure 7: Training epoch illustration

| Categories | Previous work | Our considered models | | |
|---|---|---|---|---|
| | Raychev et al. (2016a) | N2N | NT2N | NT2NT |
| One model accuracy | 83.9% | $79.4 \pm 0.2\%$ | $84.8 \pm 0.1\%$ | $84.0 \pm 0.1\%$ |
| Ensemble accuracy | | 82.3% | 87.7% | 86.2% |

Table 2: Next non-terminal prediction results

bucket, we stop adding whole programs, and start adding only the first segment of each program: when a bucket is empty, a program is chosen randomly, and its first segment is added to the bucket. We terminate the training process when all buckets are empty at the same time. That is, all programs from the first 8 epochs have been trained. This is illustrated in Figure 7.

The hidden states are initialized with $h_0, c_0$, which are two trainable vectors. The hidden states of LSTM from the previous segment are fed into the next one as input if both segments belong to the same program. Otherwise, the hidden states are reset to be $h_0, c_0$. We observe that resetting the hidden states for every new program improves the performance a lot.

We initialize all parameters in $h_0, c_0$ to be 0. All other parameters are initialized with values uniformly randomly sampled from $[-0.05, 0.05]$. For each model, we train 5 sets of parameters using different random initializations. We evaluate the ensemble of the 5 models by averaging 5 softmax outputs. In our evaluation, we find that the ensemble improves the accuracy by 1 to 3 points in general.

## 5.3 NEXT NODE PREDICTION

In this section, we present the results of our models on next node prediction, and compare them with the counterparts in Bielik et al. (2016), which is the state-of-the-art on these tasks. Therefore, we use the same testset consisting of $5.3 \times 10^7$ queries as in Bielik et al. (2016). In the following, we first report results of next non-terminal prediction and of next terminal prediction, then evaluate our considered models' performance on programs with different lengths.

**Next non-terminal prediction.** The results are presented in Table 2. From the table, we can observe that both NT2N and NT2NT can outperform Raychev et al. (2016a). In particular, an ensemble of 5 NT2N models improves Raychev et al. (2016a) by 3.8 percentage points. We also report the average accuracies of the 5 single models and the variance among them. We observe that the variance is very small, i.e., $0.1\% - 0.2\%$. This indicates that the trained models' accuracies are robust to random initialization.

Among the neural network approaches, NT2NT's performance is lower than NT2N, even given that the former is provided with more supervision. This shows that given the limited capacity of the model, it may learn to trade off non-terminal prediction performance in favor of the terminal prediction task it additionally needs to perform.

| Categories | Previous work | Our considered models | |
|---|---|---|---|
| | Raychev et al. (2016a) | NT2NT | NTN2T |
| One model accuracy | 82.9% | $76.6 \pm 0.1\%$ | $81.9 \pm 0.1\%$ |
| Overall | | 78.6% | 83.4% |

Table 3: Next terminal prediction results

| | Non-terminal | | | Terminal | |
|---|---|---|---|---|---|
| | N2N | NT2N | NT2NT | NTN2T | NT2NT |
| Top 1 accuracy | | | | | |
| Short programs (<30,000 non-terminals) | 82.3% | 87.7% | 86.2% | 83.4% | 78.6% |
| Long programs (>30,000 non-terminals) | 87.7% | 94.4% | 92.7% | 89.0% | 85.8% |
| Overall | 84.2% | 90.1% | 88.5% | 85.4% | 81.2% |
| Top 5 accuracy | | | | | |
| Short programs (<30,000 non-terminals) | 97.9% | 98.9% | 98.7% | 87.9% | 86.4% |
| Long programs (>30,000 non-terminals) | 98.8% | 99.6% | 99.4% | 91.5% | 90.5% |
| Overall | 98.2% | 99.1% | 98.9% | 89.2% | 87.8% |

Table 4: Next token prediction on programs with different lengths.

**Next terminal prediction.** The results are presented in Table 3. We observe that an ensemble of 5 NTN2T models can outperform Raychev et al. (2016a) by 0.5 points. Without the ensemble, its accuracies are around $82.1\%$, i.e., 0.8 points less than Raychev et al. (2016a). For the 5 single models, we also observe that the variance on their accuracies is also very small, i.e., $0.1\%$. On the other hand, we observe that NT2NT has much worse performance than NTN2T, i.e., by 4.8 percentage points. This shows that leveraging additional information about the parent non-terminal of the current predicting terminal can improve the performance significantly.

**Prediction accuracies on programs with different lengths.** We examine our considered models' performance over different subsets of the test set. In particular, we consider the queries in programs containing no more than 30,000 tokens, which is the same as used in Bielik et al. (2016); Raychev et al. (2016a). We also consider the rest of the queries in programs which have more than 30,000 tokens. The results are presented in Table 4.

We can observe that for both non-terminal and terminal prediction, accuracies on longer programs are higher than on shorter programs. This shows that a LSTM-based model may become more accurate when observing more code inputted by programmers.

We also report top 5 prediction accuracy. We can observe that the top 5 accuracy improves upon top 1 accuracy dramatically. This metric corresponds to the code completion scenario that an IDE may pop up a list of few (i.e., 5) candidates for users to choose from. In particular, N2N can achieve 99.1% top-5 accuracy on the non-terminal prediction task. On the other hand, NTN2T can also achieve 89.2% accuracy on the terminal prediction task. In the test set, there are 7.4% of tokens in the data whose ground truth is UNK, i.e., non-top 50,000 most frequent tokens. This means that NTN2T can predict over $89.2/(100 - 7.4)\% = 96.3\%$ of all tokens whose ground truth is not UNK. Therefore, this means that the users can choose from the popup list without typing the token manually over 96% of all time that the code completion is possible if the completion is restricted to the top 50,000 most frequent tokens in the dataset.

**The effectiveness of different UNK thresholds.** We evaluate the effectiveness of how to choose the threshold to cut for UNK terminals on the accuracy. We randomly sample 1/10 of the training dataset and the test dataset and vary the thresholds to cut for UNK terminals from 10000 to 80000. We plot the percentage of UNK terminals in both the full test set and its subset in Figure 8. We can observe that the distributions of UNK terminals are almost the same in both sets. Further, when the threshold is 10000, i.e., all terminals out of the top 10000 most frequent ones are turned into UNKs, there are more than 11% UNK queries (i.e., queries with ground truth being UNK) in the test set. When the threshold increases to 50000 or more, this number drops to 7% to 6%. The variance of the UNK queries' percentages is not large when threshold of UNK is varied from 50000 to 80000.

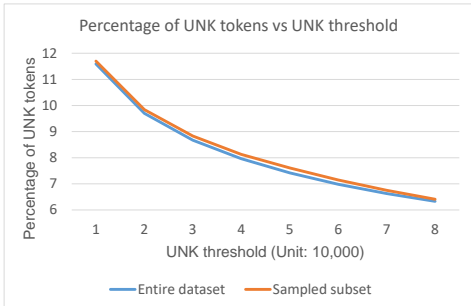
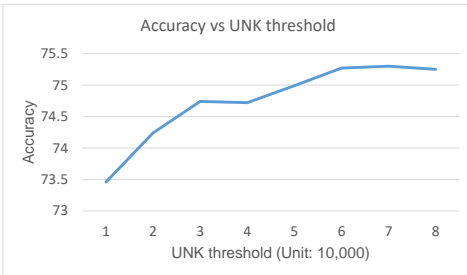

Figure 8: Percentage of UNK tokens in the entire test data and the sampled subset of the test data by varying the UNK threshold from 10000 to 80000.

Figure 9: Accuracies of different models trained over the sampled subset of training data by varying the UNK threshold from 10000 to 80000.

|  | NT2NT | N2N+NTN2T | NT2N+NTN2T |
|---|---|---|---|
| Top 1 accuracy | 73.9% | 72.0% | 77.7% |

Table 5: Predicting non-terminal and terminal together

We train one NTN2T model for each threshold, and evaluate it using the sampled test set. The accuracies of different models are plotted in Figure 9. The trend of different models' accuracies is similar to the trend of the percentage of non-UNK tokens in the test set. This is expected, since when the threshold increases the model has more chance to make correct predictions for original UNK queries. However, we observe that this is not always the case. For example, the accuracies of models trained with thresholds being 30000 and 40000 are almost the same, i.e., the difference is only 0.02%. We make similar observations among the models trained with thresholds being 60000, 70000, and 80000. Notice that we have observed above that when we train 5 models with different random initialization, the variance of the accuracies of these models is within 0.1%. Therefore, we conclude that when we increase the UNK threshold from 30000 to 40000 and from 60000 to 80000, the accuracies do not change significantly. One potential explanation is that when increasing the UNK threshold, while it has more chance to predict those otherwise UNK terminals, a model may also more likely make mistakes when it needs to choose the next terminal from more candidates.

### 5.4 JOINT PREDICTION

In this section, we evaluate different approaches to predict the next non-terminal and terminal together for the joint prediction task. In fact, NT2NT is designed for this task. Alternative approaches can predict the next non-terminal first, and then predict the next terminal based on the predicted next non-terminal. We choose NTN2T method as the second step to predict the next terminal, and we examine two different approaches as the first step to predict the next non-terminal: N2N and NT2N. Therefore, we compare three methods in total.

The top 1 accuracy results are presented in Table 5. N2N+NTN2T is less effective than NT2N+NTN2T, as expected, since when predicting the non-terminal in the first step, N2N is less effective than NT2N as we have shown in Table 4. On the other hand, NT2NT's performance is better than N2N+NTN2T, but is worse than NT2N+NTN2T.

We observe that for all these three combinations, we have

$$\Pr(\hat{T}_{k+1} = T_{k+1} \wedge \hat{N}_{k+1} = N_{k+1}) > \Pr(\hat{T}_{k+1} = T_{k+1})\Pr(\hat{N}_{k+1} = N_{k+1})$$

These facts indicate that the events of the next non-terminal and terminal being predicted correctly are not independent, but very relevant to each other instead. This is also the case for NT2NT, even though NT2NT predicts the next non-terminal and the next terminal independently conditional upon the LSTM hidden states.

|                              | NT2NT | NT2NT+D | NTN2T | NTN2T+D |
|------------------------------|-------|---------|-------|---------|
| Overall accuracy             | 81.2% | 85.1%   | 85.4% | 89.9%   |
| Accuracy on non-UNK terminals| 87.6% | 87.5%   | 92.2% | 91.8%   |
| Deny prediction rate         | 0%    | 5.2%    | 0%    | 6.1%    |

Table 6: Deny prediction results. **Top 1 accuracy** is computed as the percentage of all queries (including the ones whose ground truth is UNK) that can be predicted correctly, i.e., the prediction matches the ground truth even when the ground truth is UNK. **Accuracy on non-UNK terminals** measures the accuracy of each model on all non-UNK terminals. **Deny rate** is calculated as the percentage of all queries that a model denies prediction. **Prediction accuracy** is the top 1 accuracy over those queries that a model does not deny prediction, i.e., the prediction is not UNK.

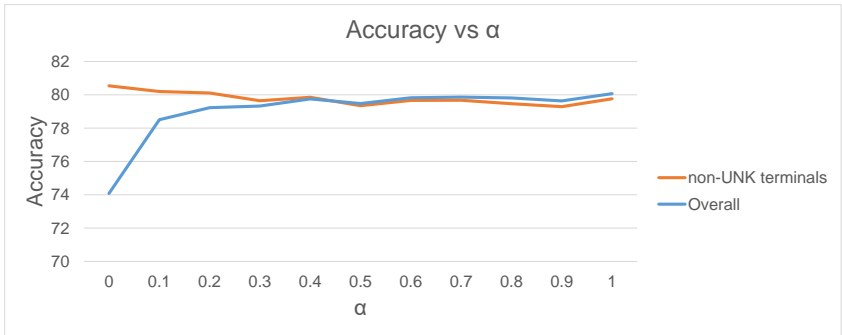

Figure 10: Overall accuracies and accuracies on non-UNK terminals by varying $\alpha$.

## 5.5 DENYING PREDICTION

We compare the models which do not deny prediction (i.e., NT2NT and NTN2T) and those which do (i.e., NT2NT+D and NTN2T+D). Results are presented in Table 6. For a reference, in the test set, there are $7.42\%$ UNK queries. We can observe that deny prediction models (i.e., +D models) have higher accuracies than the corresponding original models. This is expected. Since deny prediction models allow predicting UNK terminals, while NT2NT and NTN2T fail on all UNK queries, +D will succeed on most of them. We further evaluate the accuracy on non-UNK terminals. One may expect that since +D models may prefer to predict UNK, a standard model should have a higher accuracy on non-UNK terminals than its deny prediction counterpart. The results show that this is indeed the case, but the margin is very small, i.e., $0.1\%$ for NT2NT and $0.3\%$ for NTN2T. This means that, allowing denying prediction does not necessarily sacrifice a model's ability on predicting non-UNK terminals.

We are also interested in how frequent a +D model will deny prediction. We can observe that NTN2T+D will deny prediction for only $6.1\%$ of all queries, which is even less than the percentage of UNK queries (i.e., $7.42\%$). This shows that although we allow the model to deny prediction, it is conservative when executing this privilege. This partially explains why NTN2T+D's accuracy on non-UNK terminals is not much less than NTN2T's.

**Effectiveness of the value of $\alpha$.** We are interested in how the hyperparameter $\alpha$ in a +D model affects its accuracy. We train 11 different NTN2T+D models on the 1/10 subset of the training set, which is used above to examine the effectiveness of UNK thresholds, by varying $\alpha$ from 0.0 to 1.0. Notice that $\alpha = 0.0$, this model becomes a standard NTN2T model.

We plot both overall accuracies and accuracies on non-UNK terminals in Figure 10. We observe the same effect as above: 1) the overall accuracy for $\alpha = 1$ is $6\%$ higher than the one for $\alpha = 0$; and 2) the accuracy on non-UNK terminals for $\alpha = 1$ is less than the one for $\alpha = 0$, but the margin is not large (i.e., less than $1\%$). When we increase $\alpha$ from 0 to 0.3, we can observe that the overall accuracy steeply increases. When we further increase $\alpha$, however, the overall accuracy becomes steady. This is also the case for accuracy on non-UNK terminals. The result of this experiment

shows that how to set $\alpha$ is a trade-off between the overall accuracy and the accuracy on non-UNK terminals and how to choose $\alpha$ depends on the application.

## 5.6 RUNTIME

We evaluate our models' runtime performance. Our models are implemented in TensorFlow (Abadi et al. (2016)). We evaluate our models on a machine equipped with 16 Intel Xeon CPUs, 16 GB RAM, and a single GPU Tesla K80. All queries from the same program are processed incrementally. That is, given two queries $A, B$, if $A$ has one more node than $B$, then the LSTM outputs for $B$ will be reused for processing $A$, so that only the additional node in $A$ needs to be processed. Note that this is consistent with the practice where programs are written incrementally from beginning to end. For each model, we feed in one query at a time into the model. There are 3939 queries in total coming from randomly chosen programs. We measure the overall response latency for each query. We observe that the query response time is consistent across all queries. On average, each model takes around 16 milliseconds to respond a query on GPU, and around 33 milliseconds on CPU. Note that these numbers are from just a proof of concept implementation and we have not optimized the code. Considering that a human being usually does not type in a token within 30 milliseconds, we conclude that our approach is efficient enough for potential practical usage. We emphasize that these numbers do not directly correspond to the runtime latency when the techniques are deployed to a code completion engine, since the changes of AST serialization may not be sequential while users are programming incrementally. This analysis, however, only provides an evidence to show the feasibility of applying our approach toward a full-fledged code completion engine.

## 6 CONCLUSION

In this paper we introduce, motivate, and formalize the problem of automatic code completion. We describe LSTM-based approaches that capture parsing structure readily available in the code completion task. We introduce a simple LSTM architecture to model program context. We then explore several variants of our basic architecture for different variants of the code completion problem. We evaluate our techniques on a challenging JavaScript code completion benchmark and compare against the state-of-the-art code completion approach. We demonstrate that deep learning techniques can achieve better prediction accuracy by learning program patterns from big code. In addition, we find that using deep learning techniques, our models perform better for longer programs than for shorter ones, and when the code completion engine can pop up a list of candidates, our approach allows users to choose from the list instead of inputting the token over 96% of all time that this is possible. We also evaluate our approaches' runtime performance and demonstrate that deep code completion has the potential to run in real-time as users type. We believe that deep learning techniques can play a transformative role in helping software developers manage the growing complexity of software systems, and we see this work as a first step in that direction.

### ACKNOWLEDGMENTS

We thank the anonymous reviewers for their valuable comments. This material is based upon work partially supported by the National Science Foundation under Grant No. TWC-1409915, and a DARPA grant FA8750-15-2-0104. Any opinions, findings, and conclusions or recommendations expressed in this material are those of the author(s) and do not necessarily reflect the views of the National Science Foundation and DARPA.

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
