# Peer review of "Neural Code Completion"

_ICLR 2017 — rejected_

[Public Comment · (anonymous) · rating 4 · confidence 5 · 15 Nov 2016]
**While the overall direction is promising, there are several serious issues with the paper which affect the novelty and validity of the results.**

While the overall direction is promising, there are several serious issues with the paper which affect the novelty and validity of the results:

1. Incorrect claims about related work affecting novelty:

  - This work is not the first to explore a deep learning approach to automatic code completion: “Toward Deep Learning Software Repositories”, MSR’15 also uses deep learning for code completion, and is not cited.

  - “Code Completion with Statistical Language Models”, PLDI’14 is cited incorrectly -- it also does code completion with recurrent neural networks.

  - PHOG is independent of JavaScript -- it does representation learning and has been applied to other languages (e.g., Python, see OOPSLA’16 below). 

  - This submission is not the only one that “can automatically extract features”. Some high-precision (cited) baselines do it.

  - “Structured generative models of natural source code” is an incorrect citation. It is from ICML’14 and has more authors. It is also a log-linear model and conditions on more context than claimed in this submission.


2. Uses a non-comparable prediction task for non-terminal symbols: The type of prediction made here is simpler than the one used in PHOG and state-of-the-art (see OOPSLA’16 paper below) and thus the claimed 11 point improvement is not substantiated. In particular, in JavaScript there are 44 types of nodes. However, a PHOG and OOPSLA’16 predictions considers not only these 44 types, but also whether there are right siblings and children of a node. This is necessary for predicting tree fragments instead of a sequence of nodes. It however makes the prediction harder than the one considered here (it leads to 150+ labels, a >3x increase).


3. Not comparing to state-of-the-art: the state-of-the-art however is not the basic PHOG cited here, but “Probabilistic Model for Code with Decision Trees”, (OOPSLA 2016) which appeared before the submission deadline for ICLR’17:

[Official Review · AnonReviewer2 · rating 4 · confidence 4 · 16 Dec 2016]
**Ok paper, but not a big enough contribution**

This paper studies the problem of source code completion using neural network models. A variety of models are presented, all of which are simple variations on LSTMs, adapted to the peculiarities of the data representation chosen (code is represented as a sequence of (nonterminal, terminal) pairs with terminals being allowed to be EMPTY). Another minor tweak is the option to "deny prediction," which makes sense in the context of code completion in an IDE, as it's probably better to not make a prediction if the model is very unsure about what comes next.

Empirically, results show that performance is worse than previous work on predicting terminals but better at predicting nonterminals. However, I find the split between terminals and nonterminals to be strange, and it's not clear to me what the takeaway is. Surely a simple proxy for what we care about is how often the system is going to suggest the next token that actually appears in the code. Why not compute this and report a single number to summarize the performance?

Overall the paper is OK, but it has a flavor of "we ran LSTMs on an existing dataset". The results are OK but not amazing. There are also some issues with the writing that could be improved (see below). In total, I don't think there is a big enough contribution to warrant publication at ICLR.

Detailed comments:

* I find the NT2NT model strange, in that it predicts the nonterminal and the terminal independently conditional upon the hidden state.

* The discussion of related work needs reworking. For example, Bielik et al. does not generalize all of the works listed at the start of section 2, and the Maddison (2016) citation is wrong

[Official Review · AnonReviewer3 · rating 5 · confidence 4 · 19 Dec 2016]
**An interesting paper but only initial work about neural network based code completion.**

Pros:
  using neural network on a new domain.
Cons:
  It is not clear how it is guaranteed that the network generates syntactically correct code.

Questions, comments:
  How is the NT2N+NTN2T top 5 accuracy is computed? Maximizing the multiplied posterior probability of the two classifications?
  Were all combinations of NT2N decision with all possible NTN2T considered?

  Using UNK is obvious and should be included from the very beginning in all models, since the authors selected the size of the
  lexicon, thus limited the possible predictions.
  The question should then more likely be what is the optimal value of alpha for UNK.
  See also my previous comment on estimating and using UNK.

  Section 5.5, second paragraph, compares numbers which are not comparable.

[Official Review · AnonReviewer1 · rating 5 · confidence 4 · 20 Dec 2016]
**Great problem, too many decisions taken for granted and not explored**

This paper considers the code completion problem: given partially written source code produce a distribution over the next token or sequence of tokens. This is an interesting and important problem with relevance to industry and research. The authors propose an LSTM model that sequentially generates a depth-first traversal over an AST. Not surprisingly the results improve over previous approaches with more brittle conditioning mechanisms (Bielik et al. 2016). Still, simply augmenting previous work with LSTM-based conditioning is not enough of a contribution to justify an entire paper. Some directions that would greatly improve the contribution include: considering distinct traversal orders, does this change the predictive accuracy? Any other ways of dealing with UNK tokens? The ultimate goal of this paper is to improve code completion, and it would be great to go beyond simply neurifying previous methods.

Comments:

- Last two sentences of related work claim that other methods can only "examine a limited subset of source code". Aside from being a vague statement, it isn't accurate. The models described in Bielik et al. 2016 and Maddison & Tarlow 2014 can in principle condition on any part of the AST already generated. The difference in this work is that the LSTM can learn to condition in a flexible way that doesn't increase the complexity of the computation.

- In the denying prediction experiments, the most interesting number is the Prediction Accuracy, which is P(accurate | model doesn't predict UNK). I think it would also be interesting to see P(accurate | UNK is not ground truth). Clearly the models trained to ignore UNK losses will do worse overall, but do they do worse on non-UNK tokens?

[Author Response · Chang Liu · 17 Jan 2017]
**Methods have been improved, and paper is updated**

Dear reviewers,

We have improved our approaches and revised the paper based on the comments. We thank all reviewers for the valuable comments, and hope our work has the chance to be discussed further!

We are still working to update the paper, and some new results will be available by tomorrow.

Here is the list of all changes as of Jan-15-2017:

1) We have fine-tuned the model to achieve better performance. Now, our approaches can achieve a better performance than prior art. We have updated the following related section:
   a) Abstract & Introduction
   b) Section 5.2. Training details
   c) All results in Section 5.3 & 5.4. In 5.3, the baseline is updated to Raychev et al. (2016a).
   d) Some results in Section 5.5 are still pending. So this section is removed right now.

2) The related work section and items in the reference have been revised.

3) We have added a paragraph in Section 3.3 to explain the difference between next node prediction and next token prediction.

[Author Response · Chang Liu · 19 Jan 2017]
**A new revision to address some of reviewers' comments**

The latest revision adds the following content beyond Jan-16's version:

1) We add the results for one single models' performance (rather than an ensemble's performance) for the next non-terminal and terminal predictions in Section 5.3.
2) A new paragraph in Section 5.3 to report the effectiveness of setting the UNK threshold.
3) We add a new subsection, Section 5.5, to report the deny prediction experiments. We also add a paragraph in Sec 5.5 to examine how different choices of alpha affect a model's accuracy.

[Public Comment · (anonymous) · 22 Jan 2017]
**Technical Details of the Approach**

Given that the #’s changed by few % between the two versions, the authors should also include further technical details describing how the models were trained. In particular, several details should be easy to include such that results are reproducible, such as:

1) Is the size of the embedding the same as the hidden size (i.e., 1500)?
2) Are the embeddings pre-trained or trained simultaneously with the model? How are they pre-trained?
3) What is the motivation behind using only the first segment of a program for training after 8 epochs?
4) After 8 epochs only the first segment of every program is used for training. What is the criterion for stopping training once this process of only using the first segments has started?
5) “The last segment of a program, which may not be full, is padded with EOF tokens.” Do these EOF predictions count as correct prediction? 
6) How is the training and validation dataset selected and how many samples it contains?
7) Which dataset is used to build the vocabulary of frequent terminals used for the UNK token replacement? That is, which of the training, validation and testing datasets were used to built it?
8) What loss function is used?
9) Can you provide details on how the network parameters are randomly initialized?
10) Is regularization used? E.g. dropout? If so, what is the dropout rate?
11) Is sampled softmax used during training? If so, what is the sample size?
12) What is a typical training duration for one of the experiments from the evaluation section? Including the results for smaller models in the appendix e.g. with reduced hidden size would also be useful.

[Final Decision · Program Chairs · 06 Feb 2017]
**ICLR committee final decision**

The paper extends existing code completion methods over discrete symbols with an LSTM-based neural network. This constitutes a novel application of neural networks to this domain, but is rather incremental. Alone, I don't think this would be a bad thing as good work can be incremental and make a useful contribution, but the scores do not show an amazingly significant improvement over the baseline. There are many design choices that could be better justified empirically with the introduction of neural benchmarks or an ablation study. We encourage the authors to further refine this work and re-submit.